# Effects of Household Resource Utilization Behaviors on Giant Panda Habitat under the Background of Aging: Evidence from Sichuan Province

**DOI:** 10.3390/ijerph192215417

**Published:** 2022-11-21

**Authors:** Zhenjiang Song, Baoshu Wu, Yue Huang, Shubin Zhu, Lan Gao, Yi Li

**Affiliations:** 1Institute of New Rural Development, Jiangxi Agricultural University, Nanchang 330045, China; 2College of Economics and Management, Jiangxi Agricultural University, Nanchang 330045, China; 3School of Business Administration, Jiangxi University of Finance and Economics, Nanchang 330032, China; 4College of Economics and Management, South China Agricultural University, Guangzhou 510642, China

**Keywords:** aging labor force, resource utilization behavior, giant panda habitat, ecological niche, ecological niche overlap

## Abstract

The Giant Panda (Ailuropoda melanoleuca) is a flagship species for endangered wildlife conservation and is a specific relic species in China. Its habitat conservation has received widespread attention around the world. Since 2010, the phenomenon of an aging labor force gradually appeared within the Giant Panda Nature Reserve and its surrounding communities. Under the new labor force structure, households’ resource utilization behavior has had different characteristics, which has led an evolution in giant panda habitats. This study is based on a questionnaire and geographic data. It reveals the internal mechanisms of households’ resource utilization behavior impacting giant panda habitat patterns under the ongoing trend of labor force aging. The study shows that labor force aging has promoted rising ecological niche widths and falling ecological niche overlaps. These could drive a growth in giant panda habitat globally. From a spatial perspective, nature reserves with lower comprehensive ecological niche widths and higher ecological niche overlaps face greater conflict between conservation and development. However, the phenomenon of labor force aging mitigates these ecological conflicts to a certain extent.

## 1. Introduction

At present, the conflict between giant panda conservation and development can no longer be simply classified as a scramble for living space between a growing population and the endangered giant panda. The changes in household labor force have complicated this problem [1,2]. Following reforms that promote the opening up of Chinese society, the separation barrier between rural and urban structures has been increasingly deconstructed [3,4]. The households of the central and western regions have continued to migrate to the southeastern coast to improve their livelihoods [5,6]. After more than 40 years of rural migrant workers migrating into cities, the aging of the labor force in the central and western regions has become significant [7,8]. Agricultural laborers are now predominantly middle-aged and elderly people [9,10]. However, the aging problem is more significant when combined with livelihood stress and institutional constraints in villages around Chinese nature reserves, which are located in high mountain valleys and cold alpine region [10,11]. Therefore, the trade-off between conservation and development has been transformed into a trade-off between small-scale family farming and the expansion of giant panda habitats. The extent of this trade-off has been gradually reducing with the weakening of labor behavior. Therefore, the Chinese government has proposed an improvement of the weak links between these areas and is promoting human-giant panda coexistence [12]. At present, the habitat conservation of giant panda and its umbrella animals has entered a new stage: a harmonious development period of humanistic coordination and ecological conservation.

Many previous studies have explored the effect of household resource utilization behavior on biodiversity under the context of agricultural labor force aging. At present, the interference factors of giant panda habitat include deforestation, bamboo and its shoots cutting, fire utilization, hunting, road, herb gathering, grazing, cultivated land utilization and firewood cutting. However, the cutting of forest and bamboo have been restricted by the cuts target quota. Fire utilization and hunting have been expressly prohibited. The roads among the nature reserves have been abandoned. A series of precious Chinese herbal medicine grow in alpine meadows outside the giant panda habitat, while the proportion of herb gathering was lower. Meanwhile, grazing was usually found in alpine meadows far from the giant panda habitat and the behavior of bamboo shoots cutting was strictly limited to March and April, due to which interference was relatively small. However, cultivated land utilization and firewood cutting were closely related to the livelihood of households, which were common in Sichuan giant panda habitats. Therefore, this paper took these two resource utilization behaviors as typical representatives. Regarding the change in household resource utilization behavior, cultivated land in the edge of the forest has experienced frequent geological disasters (such as debris flow and landslides) [10,11,13], serious soil erosion [14,15] and cultivated land fragmentation [16]. These disasters have intensified the abandonment of cultivated land utilization. In addition, a large area of cultivated land presides in high mountain valleys, which is hard and steep, meaning that the aging labor force are unable to cultivate it [17,18]. The utilization rate of cultivated land has decreased with the increasingly aging labor force [19,20]. Fuelwood is cut from brushwood in high mountain valleys. However, as the labor capacity of the aging population is restricted [10,21], hydropower stations provide free electricity for households in the villages around the nature reserves and have replaced fuelwood [11,22]. Meanwhile, fuelwood utilization has been decreasing with the aging of the agricultural labor force [23,24]. Some reforms surrounding resource utilization have disappeared, such as the expanded Ecological Public-Welfare Forests [25,26], a larger Giant Panda National Park which covered important nature reserves and surrounding communities [27,28] and intensified regulations of planting behavior and firewood cutting. Under the diverse pressures of regulations [29,30], weakened labor capacity [10] and natural condition constraints [31,32], households have gradually accepted livelihood transformation and energy structure revolution.

At present, there are some research directions surrounding the effect of human activities on the giant panda habitat, such as the ecological research method and economic research method. The ecological research method sets an observation point to compare the differences between infrared image probability of giant panda [33]. It also uses the method of catch-rate-per-day and one clamp to explore the change of habitat at multiple time quantum, using capture rate as a research basis [34]. It also used FRAGSTATS to analyze the landscape pattern to explore the evolution of habitats [35]. The economic research method uses the casual model, which analyzes the effect of an economic activity (or disturbance activity) on giant panda habitats [36,37,38]. In addition, the geographical research method is also applicable. Some scholars have integrated the concepts of ecology and have taken natural geographic units as evaluation units. This could evaluate the habitat of giant pandas using a static mathematical analysis method, such as the multi-index comprehensive evaluation method, grey relational analysis or the entropy method [39]. However, a static evaluation model is unable to reflect the trajectory characteristics of giant panda habitat changes. Therefore, an analytical framework of dynamically changing spatio-temporal process should be created for the evaluation of giant panda habitats. As such, this research adopts the ecological niche model to reveal the trajectory of changes in giant panda habitats, under the context of human activity transformation, through the combination of “state” and “potential”. In the model, the ecological niche overlap accurately reflects the quantified competitive relations. The model is able to analyze the change in giant panda habitat under the context of current cultural and environmental changes in a multi-dimensional dynamic view. Therefore, the research aims to build an ecological niche model with macro-micro data interaction to explore the changes in giant panda habitat under the context of agricultural labor force aging in Sichuan giant panda nature reserves, using nine nature reserves as examples. This study could be used as a reference for the further enhancement of the habitat level of the giant panda.

## 2. Research Design

### 2.1. Interaction between Household Resource Utilization Behavior and Habitat Pattern

#### 2.1.1. Interactive Mechanisms between Household Resource Utilization Behavior and Habitat Pattern

The interactive mechanisms between household resource utilization behavior and habitat pattern occur between individual decision clustering and group decisions, and between re-clustering and overall decisions. In a rural household survey, households were considered as independent individuals, and a group consisted of households with similar decision-making behaviors. Meanwhile, a village constituted a decision-making whole. However, the clustering method was nonadditive. This was because households did not have the main direction when they made these decisions. Furthermore, these short-term changes were accumulated across time and space. Therefore, the concept of mapping was introduced into the clustering process. In the macro scale, a village was regarded as the interference source. Therefore, individuals could cluster into a whole, and villages were used as evaluation units. From the point of view of space, the research could reveal the impact of household resource utilization behavior on panda habitats. Furthermore, it allowed for the exploration of the relationship between the resource utilization behavior of the microcosmic body (household) and the mapping result of their interaction (macro-habitat landscape pattern) through multi-scale transformation (shown in Figure 1) [40].

#### 2.1.2. Construction of Individual Household Resource Utilization Decision Model

Construction of Belief Model

Belief referred to the cognition of the current system state and the estimates of the future state of agents; this definition of belief was based on a study by Wooldridge [40,41]. Consequently, the beliefs of households could be understood as the cognition of the current resource utilization behavior under the background of nature reserve regulation institution and the estimates of the future state under the context of aging labor force dominance.
*Belief*_*t*+1_ = f(*P_t_*, *Belief*_*t*_),(1)

In the equation, *Belief_t_* was the cognition of the current resource utilization behavior under the background of nature reserve regulation institution. *P_t_* was the household perception of different influencing factors in resource utilization activities.

Construction of Household Resource Utilization Decision Model

Aspiration was defined as the expected achievement state under the background of nature reserve regulation institution [40]. Therefore, households’ decisions could be calculated through the following equation:*Decision*_*t*+1_ = f(*Belief*_*t*_, *Aspiration*_*t*_),(2)

In the equation, *Decision_t_* was the decision of a household at the time *t*. *Aspiration_t_* represented the importance of household resource utilization patterns under the *Aspiration_t_*. The importance of the *k*th resource utilization pattern by the household and *j* expressed the decision of time *t* with *Belief_t_* and *Aspiration_t_* of the household at the time *t*. The model controlled for the following conditions:o The household resource utilization decisions should be mainly considered over the past five years.o There was no significant difference in the technical level of resource utilization among households.o The physical geographical environment of the households were similar.o There was no technological progress in the field of household resource utilization during the study period.o The external institutional environment remained unchanged during the study period.o There was no significant difference in labor literacy among households.o The price of resource market was stable during the study period.o The data of household resource utilization behavior came from the survey of the research group.

Therefore, a household resource utilization decision model was constructed as follows:(3)$${\mathrm{Decision}}_{\mathrm{jkt}}={\mathrm{UA}}_{\mathrm{jkt}}\;\times \;I_{\mathrm{jkt}}/\sum_{k=1}^{n}({\mathrm{UA}}_{\mathrm{jkt}}\;\times \;I_{\mathrm{jkt}}),$$

In the equation, *Decision_t_* was the importance of the *k*th resource utilization pattern by the household *j* at the time *t*. *UA_jkt_* was the area/amount/intensity of the *k*th resource utilization pattern by the household *j* at the time *t*. *I_jkt_* was the revenue of the *k*th resource utilization pattern by the household *j* at the time *t*. In the paper, *k* = 2. Therefore, the importance of the resource utilization methods was not ranked.

#### 2.1.3. Construction of Final Household Resource Utilization Decision Model

The final household resource utilization decision depended on ecological value recognition, social pressure, environmental constraints and labor literacy, under the context of aging labor force. Therefore, the decision equation for the following period could be constructed as follows:*Decision*_*jk, t*+1_ = f(*Decision*_*ikt*_|*I* = 1, 2, 3, 4),(4)

In the equation, *Decision_u_*_, *t*+1_ was the decision of the household for the next period. *Decision_it_* was the ecological value identity, social pressure, environmental constraints and labor literacy acting on household decision, respectively, at the time *t*.

In study area, there were few conversions of cropland and forest land. Land rent was relatively stable. Two rounds of reforestation had been completed. There was no government-led reforestation during the study period. The reduction in cultivated land area was the result of individual household’s decision-making behaviors. The research objects were the resource (cultivated land and fuelwood) utilization decision of households alone. This was also the case for part-time households (operating eco-tourism), part-time households (out-migration for work) and non-farm households, under the context of aging labor force dominance. Therefore, the final resource utilization decisions of individual households were mainly influenced by ecological value identity, social pressure, environmental constraints and labor literacy. These were shown as choices between different resource utilization pattern. Therefore, the above equation could be further transformed:(5)$${\mathrm{Decision}}_{\mathrm{jk},t+1}=\sum_{x=1}^{4}w_{x}{\mathrm{Decision}}_{\mathrm{jkt}},$$

In the equation, *w* was the weight coefficient of the different factors on the effect of household decision behavior. $w_{x}=a_{x}/\sum_{x=1}^{4}a_{x}$ and *w*_1_ + *w*_2_ + *w*_3_ + *w*_4_ = 1 [40,41].

### 2.2. Ecological Niche Evaluation Model

#### 2.2.1. Comprehensive Evaluation Index System of Ecological Niche

Based on the theory of “multi-dimensional super volume”, and the characteristics of the giant panda habitat level, this paper has developed a comprehensive evaluation index system with 4 dimensions and 14 measurement indexes (shown in Table 1).

#### 2.2.2. Calculating Method of Ecological Niche Width

Data Standardization

The purpose of the data standardization was to combine the units and dimensions of each index, which were nondimensionalized. The paper adopted the standard deviation method for standardization. The equation was as follows:(6)$$X_{ij}=\left|\frac{x_{ij}-\bar{x_{j}}}{\delta_{j}}\right|,$$

In the equation, *x_ij_*, *x_ij_*, $\bar{x_{j}}$, and *δ_j_* were the standardized value, original value, arithmetic mean and standard deviation of the *j*th index in *i*th community, respectively.

Calculation of index Weights

The index weights were calculated by the following equation:(7)$$V=\frac{\delta_{j}}{\bar{x_{j}}},$$
(8)$$W_{j}=\frac{V}{\sum_{j=1}^{n}V_{j}},$$

In the equation, *V* was the variable coefficient of each index. *W_j_* was the weight of the *j*th index.

Calculation of Ecological Niche Width

The calculating equation of ecological niche width was as follows:(9)$$N_{i}=\frac{W_{ik}(S_{i}+A_{i}P_{i})}{\sum_{j=1}^{n}\left(S_{j}+A_{j}P_{j}\right)},$$

In the equation, *i*, *j* = 1, 2, …, *n*. *N_i_* was the relative ecological niche in the community *i*. *S_i_* and *S_j_* were the state of communities *i* and *j*, respectively. *P_i_* and *P_j_* were the potentials of community *i* and *j*, respectively. *A_i_* and *A_j_* were the conversion coefficients of dimension. *S_i_* + *A_i_P_i_* indicated the absolute ecological niche of community *i*. *W_ik_* was the *k*-item weight of the community *i*. The value range of the ecological niche width was between 0 and 1. When the ecological niche width was closer to 1, this indicated a higher ecological level [42].

The comprehensive ecological niche width could obtain the relative ecological niche width through an arithmetic method. The equation was as follows:(10)$$M_{i}=\sum_{j=1}^{n}N_{ij}w_{j},$$

In the equation, *M_i_* was the comprehensive ecological niche in the community *i*. *N_ij_* was the relative ecological niche of the *j*-dimension in the community *i*. *w_j_* was the weight of the *j*-dimension [42].

Calculation of Ecological Niche Overlap

The ecological niche overlap reflected the relationship of ecological competition.
(11)$$O_{jk}=\frac{\sum_{i=1}^{n}P_{ij}P_{ik}}{\sqrt{\sum_{i=1}^{n}P_{ij}^{2}\sum_{i=1}^{n}P_{ik}^{2}}},$$

In the equation, *O_jk_* was the ecological niche overlap value of species *k* to species *j*. *n* was the total resource status. *P_ij_* and *P_ik_* were the percentage ownerships of resources *i* in the total resources utilized by species *j* and species *k*, respectively [42].

## 3. Study Areas and Data Sources

### 3.1. Study Areas

The study areas were the Sichuan Giant Panda Nature Reserves and their surrounding communities. According to the sampling prescription, four national natural reserves and five provincial nature reserves were sampled in this study. These were distributed across 35 communities (villages), 17 towns, 9 counties and 6 cities in Sichuan Province (shown in Figure 2). The area of giant panda habitat in the study area was 302,940.03 hm^2^, which accounted for 29.39% of total giant panda habitat area in Sichuan Province. The population of wild giant panda was 304, which accounted for 34.23% of the total wild giant panda population. Much of the youth labor force has migrated outside of the area and the aging of the labor force is significant. In addition, cultivated land has encroached on the edge of forestland. Furthermore, households still rely on fuelwood. These factors have disturbed the habitat of giant panda. Meanwhile, the researchers consulted the administrative staff of the Forestry and Grassland Administration of Sichuan Province, each giant panda nature reserve administration and each giant panda nature reserve station. The nature reserves in the samples were found across the entire scope of Giant Panda National Park. Therefore, the samples were assumed to be representative in revealing current problems faced by nature reserves.

### 3.2. Data Sources

#### 3.2.1. Survey Data and Study Sample

At present, the Wawushan Giant Panda Nature Reserve is primarily located in lots that do not belong to any individual and is a stated-owned forest. Some communities were located on the edge of the nature reserve. There were less than 10 communities with significant relevance to the nature reserves and 3–4 adjoining communities. In this paper, we adopted a stratified sampling method. Four communities were sampled in each nature reserve. Two communities were located inside the nature reserves. The community presiding inside the Tangjiahe Giant Panda Nature Reserve was formed by merging two villages. Meanwhile, two communities were located outside the nature reserves. In each community, 15–18 households were randomly sampled as respondents.

The data were obtained from household questionnaires during July 2018, October 2018, January 2019 and May 2019. A total of 557 questionnaires were distributed, and 538 valid questionnaires were obtained after the rejection of invalid questionnaires. The samples were in the experimental areas of the giant panda nature reserve and their surrounding communities, within a 5 km geographic buffer zone outside the nature reserves. Some communities were included in the scope of the Giant Panda National Park. At present, the phenomenon of labor force aging is significant in the study areas, and the resource utilization behavior of households have been constrained (as shown in Table 2).

#### 3.2.2. Survey Data on Giant Panda

The area of giant panda habitat, the population of wild giant panda, the area of nature reserve and the area of staple food bamboo were obtained from *The 3rd National Survey Report on Giant Panda in China* [43] and *The Pandas of Sichuan: The 4th survey report on giant panda in Sichuan Province* [33].

#### 3.2.3. Land Use Data

The land use data included in this paper were sourced from public Landset TM/ETM+/OLI images on Geospatial Data Cloud (http://www.gscloud.cn (accessed on 10 September 2022)), which was released by the Computer Network Information Center, Chinese Academy of Sciences. The bands, strip numbers and row numbers of two period images were used. The images were taken between June and September. This defined period made it easy to identify the information in the earth surface. The images were processed through steps such as band compositions, geometric correction, image mosaic, geometry clipmaps, supervised classification and the unification of coordinate systems. Following his, the land use patterns were classified as cultivated land, forestland, grassland, shrubland, wetland, water bodies, tundra, artificial surfaces, bareland and permanent snow and ice. These were based on the *Current Land Use Classification* (GB/T 21010-2017).

#### 3.2.4. Geographic Information Data

A series of basic vector data came from the Basic Geographic Database of China (http://www.webmap.cn/commres.do?method=result100W (accessed on 10 September 2022)) and the 4th survey on giant pandas in Sichuan Province. This data had a 1:1 million scale and the 4th survey on giant panda in Sichuan Province [41], which included roads, rivers, the points of human activity and the points of giant panda activity trace. The points of cultivated land utilization were obtained from the map of cultivated land in LUCC. The points of firewood utilization came from the survey data. A 30 m Digital Elevation Model (DEM) was obtained from the Geospatial Data Cloud, which was produced by the Chinese Academy of Sciences Computer Network Information Center (http://www.gscloud.cn/ (accessed on 10 September 2022)). Slope and slope direction were obtained based on the DEM.

## 4. Pattern of Giant Panda Habitat under the Different Age Structure of the Labor Force

### 4.1. Width of Comprehensive Ecological Niche

In this section, the measurement indexes of state were the values during 2013, and the measurement indexes of potential were the increment of each index during the period of 2013 to 2018. The dimensional transformation coefficient was one year ago. According to the methods discussed above, the ecological niche value of the giant panda habitat could gain. This included the old-age group (the aging labor force ≥55 years ago) dominating the natural resource utilization behavior of households, which was shortened to OAG and the young adult group (the young adult labor force <55 years ago) dominated the natural resource utilization behavior of households, which was shortened to YAG (shown in Figure 3).

Regarding the relative ecological niche of the level of giant panda sustainable habitat, although the level of OAG was generally higher than YAG, the difference between their ecological niche widths was small (as shown in Figure 3). As shown in the results in Figure 3A, the aging phenomena created higher levels of sustainable habitat for giant pandas. For example, the level of giant panda sustainable habitat in the Wawushan Nature Reserve was in the front of the region and displayed significant differences between the differentiation of giant panda sustainable habitat due to the labor force age structure (the ecological niche width in the OAG was 0.0078 and the ecological niche width in the YAG was 0.0076). The Xiaohegou Nature Reserve and the Heishuihe Nature Reserve displayed similar results. There were two possible reasons for this phenomenon. Firstly, labor force aging has prompted older households to shift toward less labor-intensive work, which is less disruptive to the ecological environment, such as beekeeping and white tea cultivation. These industries couple regional natural resource utilization and conservation objectives. Secondly, the aging labor force in the region either lived alone in the village, had moved to a city or lived with their children. Some young adults had left the area for employment elsewhere, meaning that the permanent population were children, women and the elderly. The impact of the size of the permanent population on divided households created miniaturization. This reduced the demand of energy in households and weakened the labor force resources, which drove households to prefer easy and cheap clean energy (such as Hydropower, and marsh gas) to adapt to the changing household structure. Therefore, the aging labor force in the region reduced the intensity of fuelwood cutting to adapt to the aging of the labor force. However, the relative ecological niche widths of the level of giant panda sustainable habitats in the Fengtongzhai Nature Reserve were lower and age had no significant effect on the sustainable habitat differentiation of the giant panda (The Ecological Niche Widths of OAG and YAG were 0.0001). There were three main reasons for this. Firstly, the industrial cluster of ecotourism was dominated by nuclear and main households, and the aging trend was not significant in the communities built after the Wenchuan Earthquake, such as David Town in Baoxing County. Secondly, the reconstructed communities were usually equipped with infrastructure for clean energy, such as marsh gas and solar energy. However, some business operations still relied on wood burning, such as for bonfire parties and heating by brazier. Therefore, households’ livelihoods still relied on firewood. Thirdly, Tibetans grew field corn to feed yak without intensive cultivation, which did not rely highly on the literacy of the labor force. Therefore, the intensity of planting behavior in the region did not fluctuate significantly with the change in the labor force structure. Therefore, the relative ecological niche widths of the level of giant panda sustainable habitat was at a relatively low level.

The widths of OAGs were generally higher than YAGs and labor force aging was more significant in the relatively higher comprehensive ecological niche widths of giant panda nature reserves (as shown in Figure 3B). According to the results, the level of the comprehensive ecological niche widths in the Wawushan Nature Reserve were the highest in the region (The ecological niche widths of the OAG in Changhe Village, Sheting Village, Shawan Village and Yanyuan Village were 0.0046, 0.0043, 0.0047 and 0.0044, respectively. The ecological niche widths of the YAG in these villages were 0.0043, 0.0041, 0.0045 and 0.0042, respectively). However, there was a large gap in the ecological niche widths between the OAG and YAG. This was attributed to the exclusivity of natural resource utilization, which only allowed these resources to be used within a given time and quantity of labor force. However, the aging labor force was weaker than the young labor force in terms of resource acquisition. This was because of their weak physiological function and ecological identity. Hence, the ecological niche widths of the OAG were higher than for the YAG. The Tangjiahe Nature Reserve, Longxi-Hongkou Nature Reserve, Xiaohegou Nature Reserve and Heishuihe Nature Reserve came second. The comprehensive ecological niche widths in Daxiangling Nature Reserve (The Ecological Niche Widths of OAG in Changsheng Village, Fazhan Village, Qiaoxi Village and Changfu Village had results of 0.0022, 0.0015, 0.0021 and 0.0022, respectively. The ecological niche widths of the YAG in these villages were 0.0022, 0.0015, 0.0020 and 0.0022, respectively). The results for the ecological niche width for the OAG in the Fengtongzhai Nature Reserve in Heping Village, Minhe Village, Qingping Village and Jiala Village were 0.0012, 0.0011, 0.0016 and 0.0007, respectively. The ecological niche widths of the YAG in these villages were 0.0011, 0.0011, 0.0016 and 0.0008, respectively. These were the most unstable. This was because Fazhan Village was close to the boundary of the nature reserve and it was also located in the southern gate of the Giant Panda National Park, leading to the development of ecotourism and overstretched natural resources. Meanwhile, ecotourism was also a leading industry in Heping Village and Qingping Village, which had a similar problem to Fazhan Village. Furthermore, Jiala Village was similar to Jiarong Tibetan village. Grazing, bonfire burning, and corn cultivation were widespread in Jiala Village, where households relied on natural resources to earn a living. Therefore, its ecological niche width was lower. In addition, the comprehensive ecological niche widths of the OAG in the Fengtongzhai Nature Reserve were slightly higher than the YAG. It also showed that the aging labor forces were weakened, which had a lesser effect on ecology. Therefore, the phenomenon of labor force aging had a regulatory effect on resource utilization decisions.

In view of the spatial pattern, nature reserves with a significantly aging labor force usually had high comprehensive ecological niche widths. The nature reserves with the best habitat were found in the Wawushan Nature Reserve in the Daxiangling Mountains. The nature reserves with relatively low comprehensive ecological niche widths were found in the Qionglai Mountains (as shown in Figure 4). Therefore, the spatial pattern of comprehensive ecological niche widths on the level of giant panda habitat further verifies the previous research results, which showed that the nature reserves and their closely related communities with a significantly aging labor force usually had relatively high ecological niche widths. Labor force aging further drove households to reduce the intensity of their resource utilization, while indirectly supporting the conservation of giant panda habitat.

### 4.2. Ecological Niche Overlap

The purpose of measuring the ecological niche overlap was to explore the competitive relationship between the communities that were close to the nature reserves (the ecological niche overlap indexes are shown in Figure 5).

The ecological niche overlap commonly existed in the OAG and the YAG. The ecological niche overlap of the YAG was slightly higher than the OAG. The values in some villages were higher than 0.800, such as in Luoyigou Village and Yinping Village in the Tangjiahe Nature Reserve. The ecological niche overlaps of Changfu Village in the Daxiangling Nature Reserve were over 0.9000. Some problems were common in these communities. Firstly, some communities were located inside the nature reserves, where there were experimental zones at an elevation of above 1 km. In these communities, households relied on the resources inside the nature reserves. Their resource utilization behavior always disturbed giant panda habitat, such as in Luoyigou Village in the Tangjiahe Nature Reserve, Lianhe community in the Longxi-Hongkou Nature Reserve and Wolong Special Administrative Region in the Wolong Nature Reserve. Secondly, some communities depended on nature reserves to operate ecotourism, which were far away from the nature reserves. However, they also had a relatively weak regulation to intensify the intensity of household resource utilization behavior where giant panda habitats would be disturbed. Thirdly, some communities relied on hydropower before the small hydropower station was shut down in 2017. These communities were Sheting Village, Shawa Village and Yanyuan Village in the Wawushan Nature Reserve. However, regional fuelwood utilization behavior significantly expanded after 2017, which caused the intensified human activities to disturb the habitat of giant pandas (as shown in Figure 5).

As shown in Figure 5, the ecological niche overlaps of the YAG were higher than the OAG in the Tangjiahe Nature Reserve. This indicates that the weakened labor force had relatively lower disturbances on giant panda habitats. The aging labor force reduced the intensity of resource utilization to restore ecology. This had a negative effect on the growth of the ecotourism economy. Therefore, the OAG had lower ecological niche overlaps. In the Wawushan Nature Reserve, the aging labor forces in Changhe Village were predominantly engaged in agroforestry to keep bees and plant white tea for the improvement of giant panda habitat level. Therefore, these aging labor forces were more sympathetic to environmentally friendly production behavior and the ecological niche overlaps of the OAG were lower than the YAG. In the Daxiangling Nature Reserve, households relied on the forestry industry in Changsheng Village. They mainly engaged in the cultivation and conservation of bamboo, willow cedar and yellow cypress. The main labor force was the OAG. Households actively supported ecological protection to improve their livelihoods and to reduce the disturbance of production activities on the ecological environment. They aimed to achieve conservation development. Therefore, it is clear that households made rational choices for sustainable livelihoods under the context of labor force aging, who relied on eco-industries.

Positive differences in ecological niche overlaps on the level of giant panda habitat were mainly found in most of regional communities under the context of labor force aging. Negative differences were mainly found in the communities which were far away from the nature reserves and had relatively frequent human activities and a relatively complex population structure, such as Wenfeng Village and Kuofeng Village in the Xiaohegou Nature Reserve, Feishui Village in the Heishuihe Nature Reserve, Jiala Village in the Fengtongzhai Nature Reserve, and Changfu Village in the Daxiangling Nature Reserve. A positive difference in the ecological niche overlap was the ratio of ecological niche overlaps of the OAG to ecological niche overlaps of the YAG, of which the data range was (0, 1). The data range of the negative differences was (1, +∞). This also showed that there were spatial differences in the effects of the aging labor force on the competing relationships between giant panda habitat levels and ecology. This further shows that the communities who lived under a special geographical environment faced resource utilization constraints and labor force weakening. This weakened the intensity of household resource utilization behavior to reduce the disturbance of giant panda habitat. Therefore, it could indirectly weaken the ecological conflicts and improve the number of giant panda sustainable habitats (as shown in Figure 6).

### 4.3. Analysis of General Tendency

This paper used the trend analysis tool, ArcGIS10.1, to identify the 3D Map with comprehensive ecological niche widths and ecological niche overlaps (as shown in Figure 7). In general, the spatial distribution of the U-shaped comprehensive ecological niche widths were similar to the inverted U-shaped ecological niche overlaps. The lowest value of ecological niche width corresponded to the highest value of ecological niche overlap. The data showed that the extreme values were in the Fengtongzhai Nature Reserve and the Wolong Nature Reserve. These were caused by human activities blocking the internal structure of the nature reserves. The data indicated that human activity destroyed the internal structure of the Fengtongzhai Nature Reserve and the Wolong Nature Reserve, which caused the low values of ecological niche widths and the high values of ecological niche overlaps. The communities distributed inside the Wolong Nature Reserve were stripe-shaped, along the central main road, which caused the isolation of giant panda habitat in the core zone and buffer zone. To develop the regional economy, some villages were defined as being outside of the Fengtongzhai Nature Reserve boundary to create a calabash-shaped nature reserve. However, the habitats on both sides of the nature reserve were isolated and faced increased disturbance from human activities. As shown in Figure 7, the differences between the ecological niche widths of the OAG and YAG were primarily in the northern region, such as in the Tangjiahe Nature Reserve, which had a relatively high level of community co-management to drive the effective transformation of the household resource utilization structure. The differences in the ecological niche overlaps were mainly in the southern region, such as in the Wawushan Nature Reserve. The livelihoods of households depended on diverse resources. Additionally, the research further confirmed that labor force aging could weaken the resource utilization behavior of households and promote sustainable habitats for giant pandas.

## 5. Conclusions and Discussion

This article analyzed the evolution of giant panda habitats in nature reserves in the Sichuan Province, from a humanistic perspective, through the household resource utilization decision model and ecological niche model, which were able to identify challenges to giant panda habitat conservation. The results showed that labor force aging has contributed to an increase in the comprehensive ecological niche widths and a decrease in the comprehensive ecological niche widths overlaps, which also somewhat alleviated ecological conflict. Furthermore, it improved the sustainable habitat levels of the giant panda.

From a global perspective, the different age-dominated resource utilization behaviors created disparate intensities of resource utilization. Household resource utilization activities created less disturbance and competitive characteristics of giant panda habitats in significant aging regions. Rising ecological niche widths and falling ecological niche overlaps appeared in these regions.

From a spatial perspective, the nature reserves in the middle region had lower comprehensive ecological niche widths and higher ecological niche overlaps with an effect on terrain structure, population distribution, boundary of nature reserves and community layout. This was the case in areas such as the Wolong Nature Reserve and the Fengtongzhai Nature Reserve. These nature reserves faced a higher conflict between conservation and development. However, the phenomenon of labor force aging could somewhat alleviate the ecological conflict, which primarily manifested in disparate ecological niche widths and ecological niche overlaps under the context of labor force weakening.

This paper aimed to evaluate the differentiation of giant panda habitats through ecological niche theory under the context of labor force aging. This is a preliminary study. In a follow-up study, we will further explore the effect of household natural resource utilization on the differentiation of giant panda habitats under different labor force structures to reveal the complex problems facing giant panda habitat conservation. The further study will be the foundation for alleviating the conflict between conservation and development.

## Figures and Tables

**Figure 1 ijerph-19-15417-f001:**
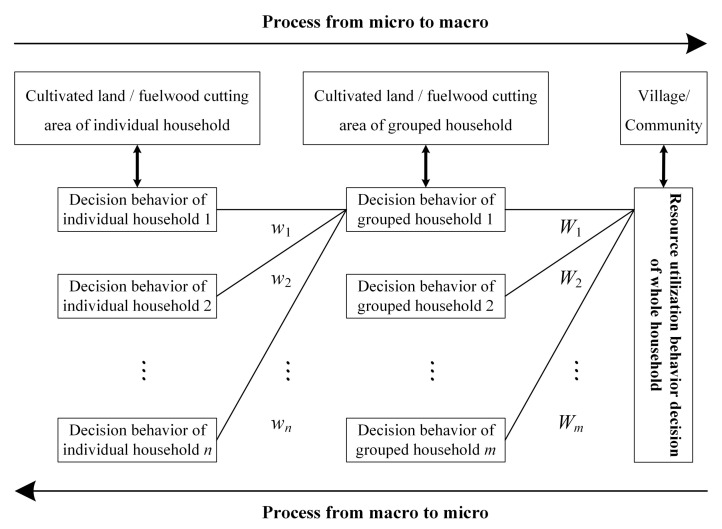
Spatial manifestation and transformation mechanism of households’ resource utilization behavior decision.

**Figure 2 ijerph-19-15417-f002:**
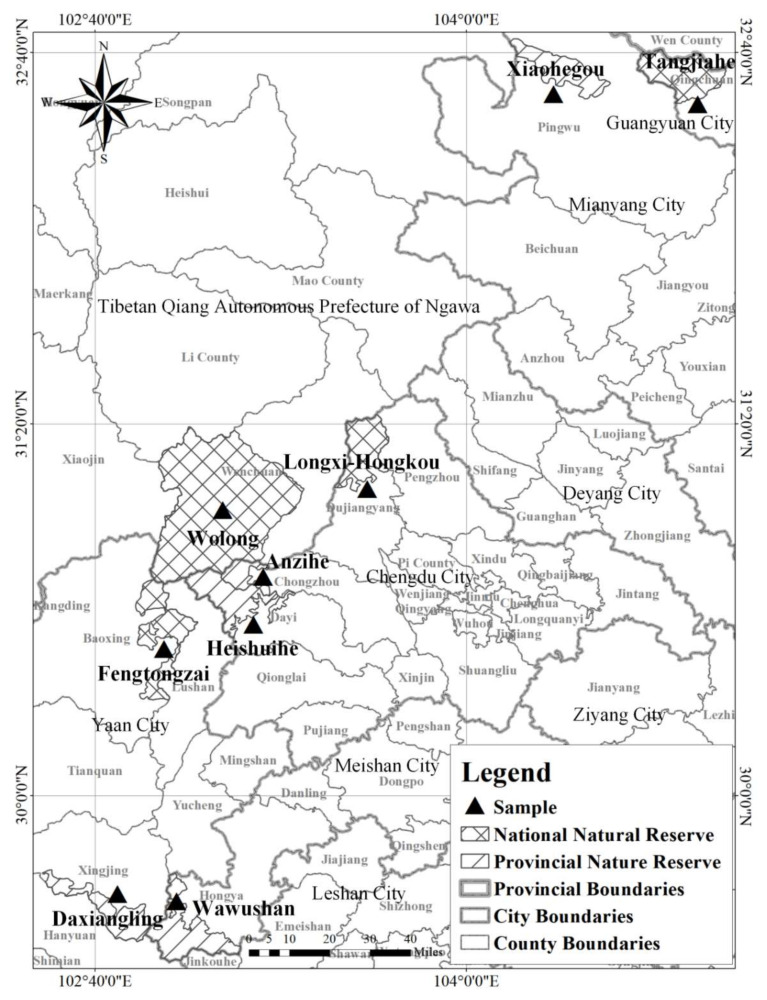
Study area.

**Figure 3 ijerph-19-15417-f003:**
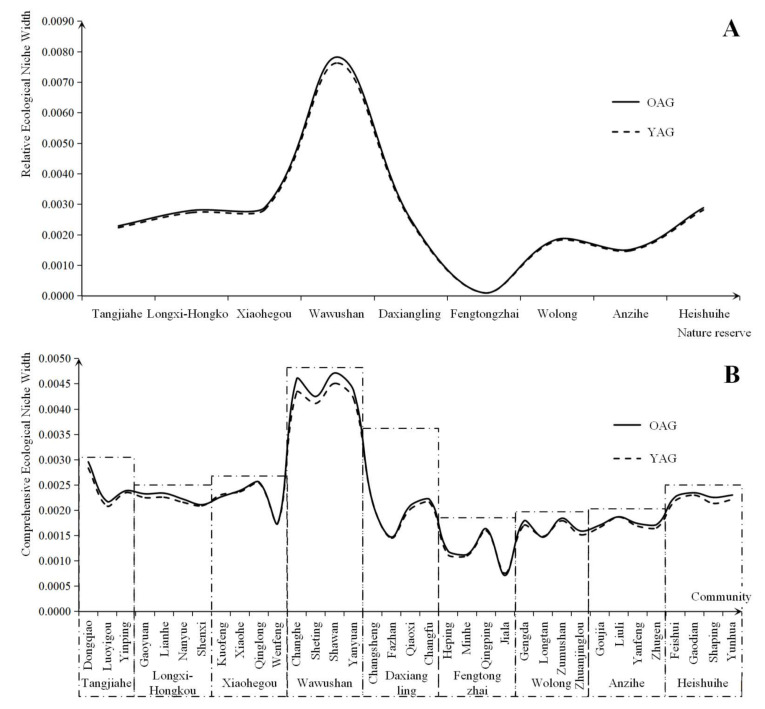
Comparison of the ecological niche on the level of giant panda habitat under the different age dominated natural resource utilization. Notes: (1) (**A**) Comparison of the relative ecological niche on the level of giant panda sustainable habitat. (**B**) Comparison of the comprehensive ecological niche on the level of giant panda sustainable habitat. (2) In the above two figures, the ecological niche widths of the OAG were generally higher than the YAG. The differences between the ecological niche widths of the OAG and YAG were smaller in the nature reserves with better giant panda habitat. (3) The level of giant panda habitat in the Daxiangling Mountain System Reserve was higher than others. The nature reserves in the Min Mountains were second. However, the giant panda habitat in the nature reserves in Qionglai Mountains faced certain risks.

**Figure 4 ijerph-19-15417-f004:**
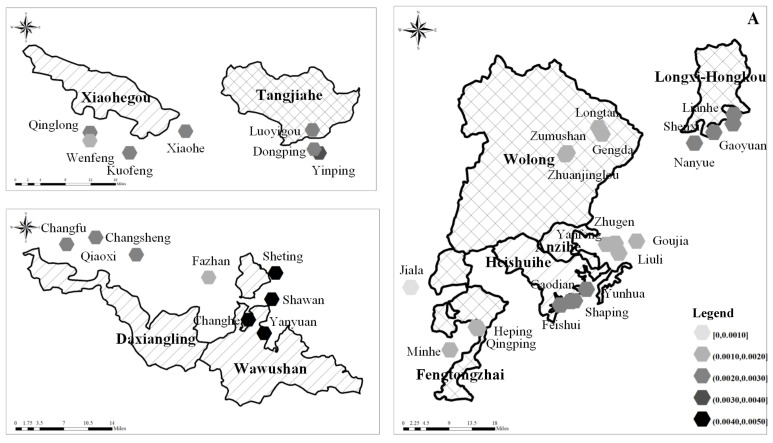
Comprehensive Ecological Niche on the level of giant panda habitat under the different age dominated natural resource utilization. Notes: (**A**) Comprehensive Ecological Niche of OAG. (**B**) Comprehensive Ecological Niche of YAG. Map Content Approval Number: CS (2019) 282.

**Figure 5 ijerph-19-15417-f005:**
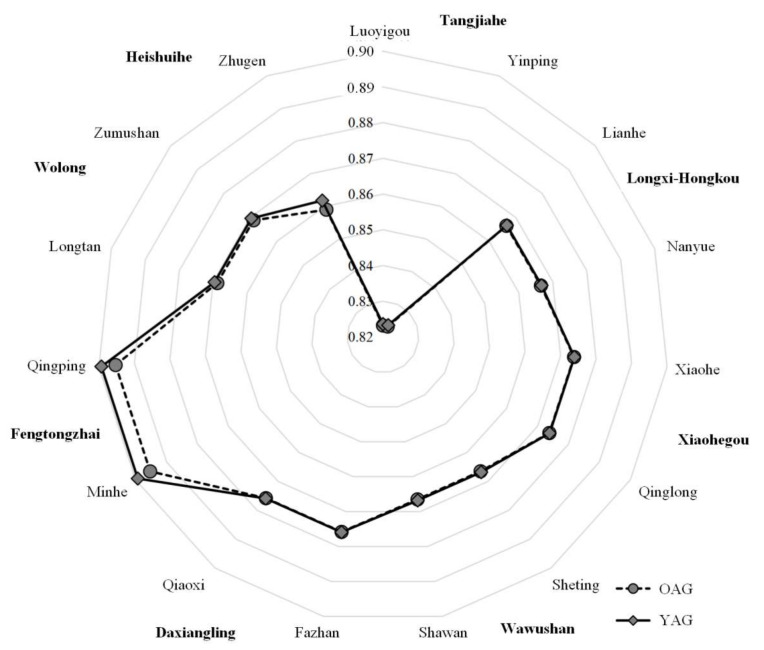
Ecological Niche Overlap Index between a part of communities and their neighboring communities.

**Figure 6 ijerph-19-15417-f006:**
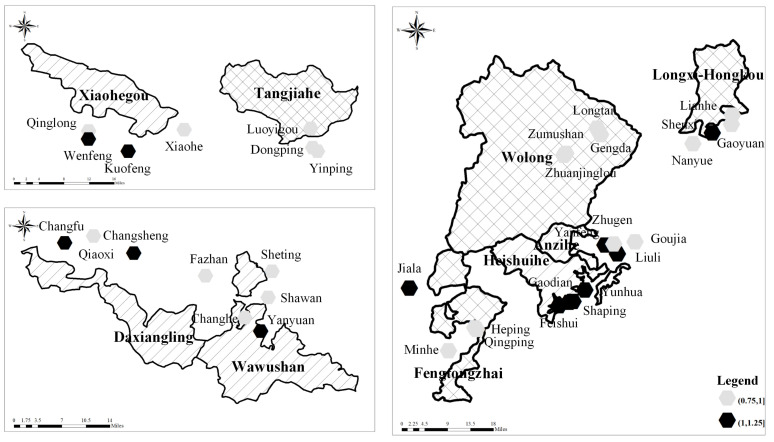
Comparison of ecological niche overlaps on the level of giant panda habitat under the labor force aging. Note: Comparison of ecological niche overlaps on the level of giant panda habitat = ecological niche overlap of OAG/Ecological Niche Overlap of YAG. Map Content Approval Number: CS (2019) 282.

**Figure 7 ijerph-19-15417-f007:**
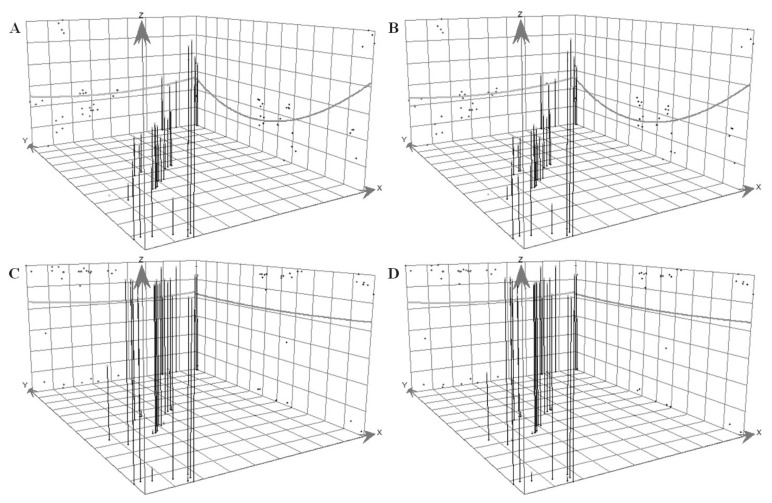
Comparison between ecological niche widths and overlaps on the level of giant panda habitat under the different age dominated natural resource utilization. Note: (**A**) Trend of ecological niche widths in OAG. (**B**) Trend of ecological niche widths in YAG. (**C**) Trend of ecological niche overlaps in OAG. (**D**) Trend of ecological niche overlaps in YAG.

**Table 1 ijerph-19-15417-t001:** Comprehensive evaluation index system of ecological niches at the habitat level of giant panda.

Target	Dimension	Measurement Index	Explanation of Indexs
Ecological Niche Width of Giant Panda Habitat	*X*_1_: Resource Utilization Strength level dimension	*x*_1_: Cultivated land utilization decision of community	Obtained by MAS model calculation, *x*_1_ ∈ [0, 1]
		*x*_2_: Fuelwood utilization decision of community	Obtained by MAS model calculation, *x*_2_ ∈ [0, 1]
	*X*_2_: Resource Utilization Support force level dimension	*x*_3_: Proportion of available cultivated land in community	*AACL*/*TACL*
		*x*_4_: Proportion of available fuelwood land/miscellaneous shrub forest in community	*AAFL*(*AMSSF*)/*TAFL*
		*x*_5_: Proportion of practitioners	*LFQH*/*TPR*
	*X*_3_: Resource Utilization Constraint level dimension	*x*_6_: Perception on intensity change of planting industry regulation	Obtained by MAS model calculation, *x*_6_ ∈ [1, 5]
		*x*_7_: Perception on intensity change of firewood cutting regulation	Obtained by MAS model calculation, *x*_7_ ∈ [1, 5]
		*x*_8_: Changing rate of cultivated land area over the last five years	Obtained by MAS model calculation, *x*_8_ ∈ [0, 1]
		*x*_9_: Changing rate of fuelwood cutting distance over the last five years	Obtained by MAS model calculation, *x*_9_ ∈ [0, 1]
		*x*_10_: Changing rate of fuelwood cutting labor-hour over the last five years	Obtained by MAS model calculation, *x*_10_ ∈ [0, 1]
	*X*_4_: Giant Panda Sustainable Habitat Level Dimension	*x*_11_: Changing rate of giant panda habitat	(*AGPH*4th − *AGPH*3rd)/*AGPH*3rd
		*x*_12_: Changing rate of wild giant panda population	(*PGPH*4th − *PGPH*3rd)/*PGPH*3rd
		*x*_13_: Area percentage of staple food bamboo	*ASFB*/*ANR*
		*x*_14_: Owned percentage of staple food bamboo in giant panda population	*ASFB*/*PGPH*4th

Note: (1) Area of available cultivated land was shortened to *AACL*. (2) Area of available fuelwood land/miscellaneous shrub forest was shortened to *AAFL*/*AMSSF*. (3) Area of giant panda habitat during the 3rd survey report on giant pandas was shortened to *AGPH*3rd. (4) Area of giant panda habitat during the 4th survey report on giant pandas was shortened to *AGPH*4th. (5) Area of staple food bamboo was shortened to *ASFB*. (6) Area of nature reserve was shortened to *ANR*. (7) Total area of cultivated land was shortened to *TACL*. (8) Total area of forestland was shortened to *TAFL*. (9) Labor force quantity of household was shortened to *LFQH*. (10) Total population of region was shortened to *TPR*. (11) Population of giant panda habitat during the 3rd survey on giant pandas was shortened to *PGPH*3rd. (12) Population of giant panda habitat during the 4th survey on giant pandas was shortened to *PGPH*4th.

**Table 2 ijerph-19-15417-t002:** Characteristic of study areas.

Variables	Explanation of Variables	Mean	Standard Deviation	Minimum	Maximum
Labor force aging	Quantity of labor force aging/Total quantity of labor force	0.289	0.327	0	1
Proportion of aging labor force in the field of planting	Quantity of aging labor force in the field of planting/Total quantity of labor force	0.478	0.287	0	1
Proportion of aging labor force in the field of fuelwood utilization	Quantity of aging labor force in the field of fuelwood utilization/Total quantity of labor force	0.455	0.281	0	1
Educational level of aging labor force	1 = Elementary school and below; 2 = Junior high school; 3 = Senior high school and above	1.271	0.887	1	3
Health level of aging labor force	1 = Serious disease and disability; 2 = Chronic disease; 3 = Health	2.385	1.439	1	3
Type of household’s livelihoods	1 = Totally dependent on agriculture; 2 = Part-time household; 3= Non-agricultural household	2.017	0.620	1	3
Soil quality	1 = Poor; 2 = General; 3 = Well	1.704	1.120	1	3
Degree of cultivated land fragmentation	(Quantity of cultivated land plots − 1)/(Total area of cultivated land/Area of minimum cultivated land plot in the study area)	0.733	1.507	0	1
Slope of Fuelwood cutting destination	1 = Gentle; 2 = Steeper; 3 = Extremely steep	2.617	1.241	1	3
Restriction of fuelwood cutting	1 = Existence; 0 = Inexistence	0.684	0.474	0	1
Mean distance of fuelwood cutting destination	Kilometer	4.891	4.247	0	16

Note: (1) From the perspective of households’ resources utilization, the proportion of aging labor force in the field of planting and fuelwood utilization were 47.80% and 45.50%. Therefore, aging labor force has become the main body of resource utilization activities. (2) From the perspective of labor force literacy, lower educational level and sub-health status was common among aging labor forces, whose mean values were 1.271 and 2.385. (3) From the perspective of resource utilization constraints, the trend towards non-agriculture (2.017) has been obvious in the study area. Meanwhile, less fertile soil (1.704) and fragmented cultivated land (0.733) could restrict the planting behavior of households. Extremely steep topography (2.617), strongly restriction of fuelwood cutting (0.684), and far distance of fuelwood cutting destination (4.891) could restrict the fuelwood utilization behavior of households.

## Data Availability

Survey data were obtained from household questionnaires during July 2018, October 2018, January 2019 and May 2019. Survey data on giant panda were obtained from *The 3rd National Survey Report on Giant Panda in China* and *The Pandas of Sichuan: The 4th survey report on giant panda in Sichuan Province*. The land use data of this paper was sourced from public Landset TM/ETM+/OLI images on Geospatial Data Cloud (http://www.gscloud.cn (accessed on 10 September 2022)), which was released by Computer Network Information Center, Chinese Academy of Sciences. A series of basic vector data came from the Basic Geographic Database of China (http://www.webmap.cn/commres.do?method=result100W (accessed on 10 September 2022)) and the 4th survey on giant pandas in Sichuan Province.

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
