# Peer review of "Effects of Household Resource Utilization Behaviors on Giant Panda Habitat under the Background of Aging: Evidence from Sichuan Province"

_ijerph, 2022, doi:10.3390/ijerph192215417_

Round 1

Reviewer 1 Report

This is a very well done, interesting study. The results are not dramatic but are interesting, and will be surprising to those not aware of recent changes in China, here, in particular, migration to cities with consequent aging of rural workforce. The research used detailed methods to study the effects of this aging on panda conservation. It is evidently reducing pressure on the resource base of the panda reserves. As the authors point out, this is a preliminary study, so we are not yet aware of how much effect this aging will have. In any case, this is an interesting project. So far as I am aware, it is unique; previous studies of the aging rural workforce that I have seen have dealt with its effect on agriculture, rural society, and rural households, not with panda conservation.

The English is good, but there are some minor problems, such as lack of "the" in places, and a mysterious "the through" in line 516.

Author Response

We are grateful to the editor and reviewers for their positive and constructive comments and criticisms concerning our manuscript, Effects of Household Resource Utilization Behaviors on Giant Panda Habitat Under the Background of Aging: Evidence from Sichuan Province(ijerph-1963015). These comments and criticisms are very helpful for revising and improving our paper. We have made necessary corrections and changes in response to them. We hope that you will find our revised manuscript is acceptable for publication. Of course, we will make additionally changes if there remain unaddressed or inadequately addressed comments.

Reviewer #1:

Thanks for your observation and suggestion. Based on the review comments, we have: (1) improved the quality of language; (2) checked the information in the paper.

Reviewer 2 Report

I have to say that this manuscript has not been appropriately prepared and should not be sent for peer review. The authors need to learn some basic principles in presenting a study. All my comments are in the attached PDF. 

Author Response

We are grateful to the editor and reviewers for their positive and constructive comments and criticisms concerning our manuscript, Effects of Household Resource Utilization Behaviors on Giant Panda Habitat Under the Background of Aging: Evidence from Sichuan Province(ijerph-1963015). These comments and criticisms are very helpful for revising and improving our paper. We have made necessary corrections and changes in response to them. We hope that you will find our revised manuscript is acceptable for publication. Of course, we will make additionally changes if there remain unaddressed or inadequately addressed comments.

Reviewer #2:

  1. checked the information in the paper.

Thanks for your observation and suggestion. I have checked and changed a series of information in the paper. And some points has used in previous literature, such as Labor Force Aging has been used in some paper, such as Zhou et al., 2015 and Prenzel et al., 2021. And households’ resource utilization behavior has been used in some paper, such as Duan et al., 2022 and Hu et al., 2018.

The word of scramble means that to push, fight or compete with others in order to get or to reach sth., which has shown the intense degree of the conflict between giant panda conservation and development.

Although the policy of rural migrant worker has been published in No. 1 central document of 1985, but a large number of rural migrant workers have worked in cities in 1980. Therefore, the time is more than 40 years.

Umbrella animal is biological terminology. It means that flagship species are protected, while other species are also protected.

In Sichuan Province, hydropower stations provide free electricity for households in the villages, and the extra electricity. And hydropower stations are able to sell back surplus electricity at premium prices to the national grid.

Meanwhile, I have changed. “At present, there are some research directions surrounding the effect of human activities on giant panda habitat, such as the ecological research method and economic research method.”

The ecological research method focus on fixed point measurement through infrared camera in giant panda habitat. And research method is compare the differences between infrared images probability on giant panda.

The habitat level is the classification of habitat environment.

In GlobeLand30, LUCC has been classified as cultivated land, forest, grassland, shrubland, tundra, artificial surfaces, water bodies, wetland, permanent snow and ice, and bareland. Therefore, we use cultivated land in this paper.

  1. Some problems in Interactive Mechanisms Between Household Resource Utilization Behavior and Habitat Pattern.

We appreciate these valuable comments. Individual decision clustering is the set of decision behavior of individual households. And group decisions is the clustering of individual decisions. Because a village is regarded as a source of interference, which series of resource utilization behavior could disturb the habitat of giant panda. At the macro-scale, the village as a whole existed, which is the evaluation unit of interference. And we has changed the formulation of last four sentences. And we have changed these expressions. “In the macro scale, a village was regarded as interference source. Therefore, individuals could cluster into a whole, and villages were used as evaluation units. From the space point of views, the research could reveal the impact of household resource utilization behavior on panda habitat. Furthermore, it allowed the exploration of the relationship between the resource utilization behavior of thr microcosmic body (household) and the mapping result of their interaction (macro-habitat landscape pattern) through multi-scale transformation (shown in Figure 1) [41].” And Figure 1 has changed.

In the part of Construction of Belief Model, we have changed the expressions. “Thereout, the beliefs of households could be understood as the cognition on the current resource utilization behavior under the background of nature reserve regulation institution and the estimates on future state under the context of aging labor force dominance.” “In the equation, Belieft was the cognition on the current resource utilization behavior under the background of nature reserve regulation institution.”

In the part of Construction of Household Resource Utilization Intention Model, we have changed the expressions. The aspiration was the expected achievement state under the background of nature reserve regulation institution [41].

  1. Checked the information in the paper.

Thank you for pointing this out. I have checked the information. And these figures have changed.

Reviewer 3 Report

1. Please clarify the literature basis of construction of individual household resource utilization decision model and final household resource utilization decision intention model. And explain those questions in the survey that are used for analysis.

2. What are the demographic characteristics of the sample? 

3. The OAG and YAG were not clearly defined from the age statistics.

4. The construction of models are "intentions" and how it is linked with household resource utilization "behaviors"?

5. Please clarify what household resource utilization behaviors contain in detail. And add more analysis details about why household resource utilization behaviors are linked with the changes of giant panda habitat patterns. How to exclude other explanatory variables?

Author Response

We are grateful to the editor and reviewers for their positive and constructive comments and criticisms concerning our manuscript, Effects of Household Resource Utilization Behaviors on Giant Panda Habitat Under the Background of Aging: Evidence from Sichuan Province(ijerph-1963015). These comments and criticisms are very helpful for revising and improving our paper. We have made necessary corrections and changes in response to them. We hope that you will find our revised manuscript is acceptable for publication. Of course, we will make additionally changes if there remain unaddressed or inadequately addressed comments.

Reviewer #3:

  1. Please clarify the literature basis of construction of individual household resource utilization decision model and final household resource utilization decision intention model. And explain those questions in the survey that are used for analysis.

Thanks for your observation and suggestion. The construction of individual household resource utilization decision model and final household resource utilization decision intention model were based on the literature of Chen et al. (2011) and Wooldridge (2002).

  1. 2. What are the demographic characteristics of the sample?

We appreciate these valuable comments. We have added the demographic characteristics.

“At present, the phenomenon of Labor force aging was significant in the study areas, and resource utilization behavior of households have been constrained (as shown in Table 2).”

Table 2. Characteristic of study areas.

Variables

Explanation of Variables

Mean

Standard deviation

Minimum

Maximum

Labor force aging

Quantity of labor force aging / Total quantity of labor force

0.289

0.327

0

1

Proportion of aging labor force in the field of planting

Quantity of aging labor force in the field of planting / Total quantity of labor force

0.478

0.287

0

1

Proportion of aging labor force in the field of fuelwood utilization

Quantity of aging labor force in the field of fuelwood utilization / Total quantity of labor force

0.455

0.281

0

1

Educational level of aging labor force

1 = Elementary school and below; 2 = Junior high school; 3 = Senior high school and above

1.271

0.887

1

3

Health level of aging labor force

1 = Serious disease and disability; 2 = Chronic disease; 3 = Health

2.385

1.439

1

3

Type of household's livelihoods

1 = Totally dependent on agriculture; 2 = Part-time household; 3= Non-agricultural household

2.017

0.620

1

3

Soil quality

1 = Poor; 2 = General; 3 = Well

1.704

1.120

1

3

Degree of cultivated land fragmentation

(Quantity of cultivated land plots − 1) / (Total area of cultivated land / Area of minimum cultivated land plot in the study area)

0.733

1.507

0

1

Slope of Fuelwood cutting destination

1 = Gentle; 2 = Steeper; 3 = Extremely steep

2.617

1.241

1

3

Restriction of fuelwood cutting

1 = Existence; 0 = Inexistence

0.684

0.474

0

1

Mean distance of fuelwood cutting destination

Kilometer

4.891

4.247

0

16

Note: â—‹ From the perspective of households’ resources utilization, the proportion of aging labor force in the field of planting and fuelwood utilization were 47.80% and 45.50%. Therefore, aging labor force has became the main body of resource utilization activities.

â—‹ From the perspective of labor force literacy, lower educational level and sub-health status was common among aging labor forces, whose mean values were 1.271 and 2.385.

â—‹ From the perspective of resource utilization constraints, the trend towards non-agriculture (2.017) has been obvious in the study area. Meanwhile, less fertile soil (1.704) and fragmented cultivated land (0.733) could restrict the planting behavior of households. Extremely steep topography (2.617), strongly restriction of fuelwood cutting (0.684), and far distance of fuelwood cutting destination (4.891) could restrict the fuelwood utilization behavior of households.

  1. 3. The OAG and YAG were not clearly defined from the age statistics.

Thank you for pointing this out. “The aging labor force (≥ 55 years ago) dominated the natural resource utilization behavior of households, which was shortened to OAG.” “The young adult labor force (< 55 years ago) dominated the natural resource utilization behavior of households, which was shortened to YAG.”

  1. 4. The construction of models are "intentions" and how it is linked with household resource utilization "behaviors"?

We appreciate these valuable comments. Individual decision clustering is the set of decision behavior of individual households. And group decisions is the clustering of individual decisions. Because a village is regarded as a source of interference, which series of resource utilization behavior could disturb the habitat of giant panda. At the macro-scale, the village as a whole existed, which is the evaluation unit of interference. And we has changed the formulation of last four sentences. And we have changed these expressions. “In the macro scale, a village was regarded as interference source. Therefore, individuals could cluster into a whole, and villages were used as evaluation units. From the space point of views, the research could reveal the impact of household resource utilization behavior on panda habitat. Furthermore, it allowed the exploration of the relationship between the resource utilization behavior of thr microcosmic body (household) and the mapping result of their interaction (macro-habitat landscape pattern) through multi-scale transformation (shown in Figure 1) [41].” And Figure 1 has changed.

In the part of Construction of Belief Model, we have changed the expressions. “Thereout, the beliefs of households could be understood as the cognition on the current resource utilization behavior under the background of nature reserve regulation institution and the estimates on future state under the context of aging labor force dominance.” “In the equation, Belieft was the cognition on the current resource utilization behavior under the background of nature reserve regulation institution.”

In the part of Construction of Household Resource Utilization Intention Model, we have changed the expressions. The aspiration was the expected achievement state under the background of nature reserve regulation institution [41].

  1. 5. Please clarify what household resource utilization behaviors contain in detail. And add more analysis details about why household resource utilization behaviors are linked with the changes of giant panda habitat patterns. How to exclude other explanatory variables?

Thanks for your observation and suggestion. We have explained the problem in Introduction.

“At present, the interference factors of giant panda habitat included deforestation, bamboo and its shoots cutting, fire utilization, hunting, road, herb gathering, grazing, cultivated land utilization and firewood cutting. However, the cutting of forest and bamboo have been restricted by the cuts target quota. Fire utilization and hunting have been expressly prohibited. The roads among the nature reserves have been abandoned. A series of  precious Chinese herbal medicine grew in alpine meadows outside the giant panda habitat, while the proportion of herb gathering was lower. Meanwhile, grazing was usually found in alpine meadows far from giant panda habitat. And the behavior of bamboo shoots cutting was strictly limited to March and April with regular time, which interference was relatively small. But ultivated land utilization and firewood cutting were closely related to the livelihood of households, which were common in Sichuan giant panda habitats. Therefore, this paper took these two resource utilization behaviors as typical representatives.”

Round 2

Reviewer 3 Report

More explanations are still needed for the gap between "intentions" and "behaviors"(i.e., intentions do not necessarily lead to behaviors), for the Question 4. The construction of models are "intentions" and how it is linked with household resource utilization "behaviors"?

After the intention model is built, and then how you integrate it to your analysis? Please explain in detail the logics behind, but not only quote the "text".

Author Response

We appreciate these valuable comments. In this paper, Intention as willingness is not accurate. Due to the limit of reference, this vocabulary should be corrected to decision. Because the paper aims to build Individual Household Resource Utilization Decision Model. Of course, there is a gap between intentions and behaviors. And intentions don’t necessarily translate into behaviors. In the nature reserves and its surrounding communities, intentions and behaviors have different effects on the habitat of giant panda. Therefore, we have studied this problem in further study. In next paper, we have quantified the gap between intention and behavior, to identify the difference of disturbance on giant panda habitat. Meanwhile, we will explore the mitigation mechanism.
